# Need for affect, need for cognition, and the desire for independence

**Joan Barceló** [ID] *

Division of Social Science, New York University - Abu Dhabi, Abu Dhabi, United Arab Emirates

* joan.barcelo@nyu.edu

**Data Availability Statement:** The data underlying the results presented in the study are available from Harvard Dataverse (https://doi.org/10.7910/DVN/1POM33).

**Funding:** The author(s) received no specific funding for this work.

## Abstract

The last decade has witnessed a significant rise in European secessionist movements leading to considerable political turmoil (e.g., protest, repression, imprisonment of political leaders). While scholars have identified a number of economic and political factors that influence the support for secessionist movements, fewer studies have focused on its psychological roots. Using evidence from Catalonia, this paper investigates the role of two fundamental individual traits, Need for Affect (NFA) and Need for Cognition (NFC), in influencing the support for Catalan independence. It analyzes a large representative sample of adult Catalans in 2013, during the peak of the secessionist movement, to examine the influence of NFA and NFC, and their interaction, on the intention to vote in favor of seceding from Spain. Results indicate that individuals who have high levels in NFA and those who have high levels of NFA and low levels of NFC are more likely to support independence. In other words, individuals who have low levels of both NFA and NFC have the lowest support for independence. In conclusion, these findings highlight the importance of considering individual differences in psychological motives in order to fully understand support for secessionist movements.

## Introduction

Secessions are major political events that shape our world map once in a while. The dramatic breakup of states such as the Soviet Union, Yugoslavia, and Czechoslovakia, along with successful separatist cases like Eritrea from Ethiopia, Timor Leste from Indonesia, and Montenegro and Kosovo from Serbia, following the end of the Cold War, has substantially increased the number of countries. Additionally, the last few decades have also witnessed the strengthening of movements for independence in many advanced democracies such as Belgium (Flanders), Canada (Quebec), Spain (Basque Country and Catalonia), the United Kingdom (Scotland), Italy (Padania), among others. Some of these regions have even held self-determination referendums such as Quebec in 1980 and 1995 and Scotland in 2014. Democratization and an increasing international economic and political integration may be the driving force of the revival of self-determination movements [1, 2]. This has led to a growing interest in understanding the factors that shape the formation of new countries and the redrawing of state boundaries.

**Competing interests:** The authors have declared that no competing interests exist.

Researchers across the social sciences have devoted much attention to accounting for the macro-determinants of secessionist movements; that is, why we observe territorial minorities claiming independence in some countries and not in others. Most scholars agree that territorial minorities are more likely to advocate for secession if they enjoy an economically advantaged position vis-à-vis the rest of the state [3, 4], have high levels of regional economic inequality, a significant share of the country's population [5], a decentralized party system, and a cultural singularity (e.g., a distinct language) [6–9].

Beyond the study of macro-determinants, scholars have also paid considerable attention to the micro-level determinants of the support for independence movements; that is, why some individuals support independence and not others in a territory with a relevant secessionist movement. On the one hand, some scholars have studied the significant role of regional elites and local media in stirring up secessionist demands [10–14]. On the other hand, another set of scholars has paid attention to the individual-level dynamics from a sociological perspective. In this literature, the focus has been nearly exclusively on the socio-political variables that influence the support for independence. Primary socio-demographic variables such as age, gender, or birthplace; secondary socialization factors such as social class, education, language, or sources of political information; and socio-political factors such as national identity, ideological self-placement, and economic preferences are common predictors in most empirical models [15–22].

An emerging body of scholarship has begun to consider the potential impact of basic individual differences on the desire for independence. These include systematic evaluations of the impact of moral sentiments [23], personality traits [24], and risk preferences [25–29]. Using the rise of the secessionists' demands in Catalonia, this paper contributes to this literature by exploring the role of basic psychological motives, the psychological Need for Affect (NFA) and Need for Cognition (NFC), on individuals' behavioral intention to support Catalan independence. As the Catalan secessionist movement significantly strengthened in the early 2010s in Catalonia, regional elites actively stirred up secessionist demands at the same time as the public space became monopolized by the movement through emotion-inducing activities (i.e., symbols, demonstrations).

During this time, I argue that individuals who score high in the psychological motive of Need for Affect, which describes a general tendency to approach emotions and emotional-inducing situations, would be more likely to support the pro-independence movement. When high levels of NFA are combined with an absence of NFC (the so-called "feelers"), individuals have a tendency to strictly follow their emotional reaction in response to empirical facts. Therefore, the effects of NFA on secessionism should be strongest when NFA is combined with an absence of NFC. A large-scale public opinion survey conducted in 2013—the peak of the secessionist movement— reveals evidence in favor of these theoretical expectations. These results suggest that psychological motives are fundamental determinants to explain the support for self-determination movements in advanced democracies.

## Context: Catalan independence movement

Catalan governments have attempted to expand the scope of the regional self-government ever since the transition to democracy in 1978. Regional governments had been rather successful at achieving greater powers in the first two decades after the transition. During this period, support for secessionism was remarkably rare. A turning point occurred in the early 2000s as the central government became reluctant to deepen the decentralization process. For the government of José María Aznar (1996–2004), the constitutional development of the Spanish "state of

the autonomies" had been completed, and the final transfer of powers to the regions in the mid-1990s marked the end of the decentralization process.

Given that the political framework was deemed insufficient by Catalan nationalist parties, they initiated a formal process for the Reform of the Statute of Autonomy of Catalonia to move toward a new stage in the devolution process. After a tedious and contentious process of elaborating, negotiating, and approving a new Statute of Autonomy for Catalonia, the Constitutional Court made the controversial decision to curtail key aspects of the reform. Some Catalan parties and regional leaders framed the Constitutional Court's controversial decision as the glass ceiling of self-rule within Spain. Some authors regard this as the key moment that triggered the subsequent growth of Catalan's support for independence [30, 31].

Fig 1 shows the evolution of support for independence according to a survey measure taken from the *Centre d'Estudis d'Opinió*. The support for Catalan independence more than doubled in three years. Between 2010 and 2013, it grew from around 20% to almost 50%. The abrupt increase of the willingness to secede between 2010–2013 is largely connected to the regional elections of 2012 that were called by a nationalist-lead Government that had changed from a decades-long moderate politics centered on enhancing devolution from Spain to a direct secession pathway. After 2014, the support for Catalan independence slightly declined, and it has plateaued at around 35%-40% until today. Therefore, this project explores the role of psychological motivations, NFA and NFC, in explaining the surge of the support for secessionism in the unique time —mid-2013— in which the secessionist movement was booming. In other words, this project can provide us with information about what are the psychological needs that individuals can fulfill by joining an expanding secessionist movement.

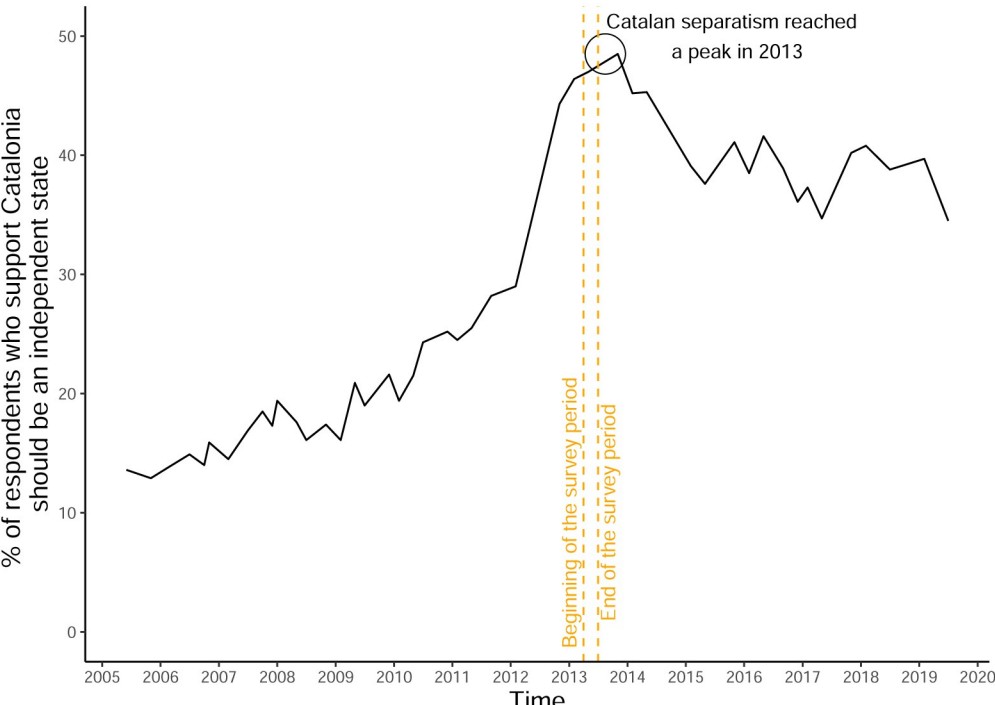

**Fig 1. Evolution of the support for Catalan independence (2006–2019).** The question from the *Centre d'Estudis d'Opinió* (CEO) used to create the figure has the following wording: "Which kind of political entity should Catalonia be with respect to Spain? Independent state/ Federal state within Spain/Autonomous community within Spain/Region within Spain." This question is used—rather than the direct question on whether respondents support independence or not—because it has been asked for a longer time span by the CEO.

The rise in the desire for Catalan independence in the period 2012–2013 had direct consequences in the political realm. The percentage of respondents who explicitly support that Catalonia should be an independent state has remained stable around 35%-45% since 2013, which is about double the support than it received before 2012. In this decade, regional political parties and institutions have taken a stronger pro-independence stance, which has led to several critical events, including numerous mass-scale demonstrations (2012-), the organization of illegal self-determination referendums (2014 and 2017), the imprisonment of Catalan political and social leaders (2017-), and a protest movement (2013-) with significant levels of violence and rioting activity (2019) such as the prolonged rioting in Barcelona after the Supreme Court of Spain sentenced nine Catalan independence leaders in October 2019. Additionally, some evidence suggests that the the Spanish governments' repressive actions might have also spurred the conflict after 2017 [32–35]. Overall, the intensity of the secessionist conflict has led to increased social polarization in the region [36].

## Psychological motivations and the desire for independence

Extant literature has focused on the origins of the support for self-determination movements in advanced democracies [20, 21, 37–41]. Even though there is an important variety of scholarly work, the *identity model* and the *instrumental model* are the two major theoretical models that explain individual differences in the support for secessionist movements. The identity model supports the thesis that most individuals generate affective attachments to territorial entities (e.g., countries, regions) through early socialization processes [38]. The type and intensity of these attachments become later reflected in peoples' national identities [37]. Analogous to partisanship in the American literature [42], national identity is similarly viewed as an "unmoved mover" in which individuals obtain some utility by orienting their political attitudes and behavior in a way that fits with their national identification [16]. In this regard, people's strong identification with the region—as opposed to the state—fundamentally influences people's likelihood to support a nationalist or independence movement [21].

Contrary to this approach, the instrumental model suggests that the support for self-determination movements is the result of a cost-benefit analysis of the potential consequences of reaching full sovereignty. On the one hand, some individuals may believe that independence could grant greater fiscal capacity for the region—especially if the region is developed relative to the rest of the country—and, thus, a chance for strengthening the welfare state and implement policies that promote economic growth [16, 20]. On the other hand, the cost-benefit calculation of supporting a self-determination movement factors in the potential negative shock of independence (the prospects of economic chaos and the exclusion from important international alliances such as the NATO or the EU, although there is also contrary evidence on this effect [1, 43].). Overall, the instrumental model predicts that positive (negative) beliefs about the impact of independence on individual outcomes would predict a greater (lower) chance to support independence.

The identity and the instrumental models generate different predictions about how individuals who live in areas with a relevant secessionist movement process and update political information about independence. The identity model suggests that national identities are socially created constructs that, once forged, remain stable over time, and shape the interpretation of salient facts around the topic of the secession. The instrumental model proposes that there is a strong association between factual beliefs about the consequences of the secession and the intention to support it. In this regard, facts that change the prospects of benefiting from secession would lead to variation in individuals' support for independence. From this perspective, information leads to updating attitudes and behavior in a rational manner.

Survey and experimental work have been extensively used to adjudicate between the identity and the instrumental model [20, 39, 40]. As it happened with the study of partisanship [44], researchers have yet to agree on which of the two models best describes the intention to support secession. While most scholars have so far implicitly assumed that citizens evaluate information through the same cognitive process, psychologists seem to support the alternative notion that individuals possess different psychological motivations when processing information [45]. Following this path, this paper evaluates the extent to which two major psychological motivations—Need for Affect and Need for Cognition—may help us explain support for the self-determination movements.

Psychological motivations are deeply involved in the process of belief formation and have relevant implications for political behavior [46]. Motivations are referred to as directional when they are related to specific conclusions and oriented towards the actual conclusions drawn. However, here I always refer to non-directional motivations, which do not influence the final conclusions but, instead, the cognitive processes used to reach those conclusions [46]. Among these, two non-directional motivations that may be important in politics are Need for Affect (NFA) and Need for Cognition (NFC). While NFA and NFC are non-directional psychological motivations, NFA and NFC can be predictors of preferences and behaviors contingent on a specific sociological context. This interpretation fits with the extant literature showing that how you process information (cognitive assessments or affective evaluations) affects what you think about an issue and shapes how you behave. Obviously though, individuals' stimuli and experiences is contingent on the context. For an illustrative literature review of the direct effects of the NFA or NFC on attitudinal and behavioral outcomes [47].

NFA describes the general motivation to approach or avoid emotions or emotional-inducing situations and includes the belief that emotions are useful for shaping judgments and behavior [48]. Those high in NFA tend to hold attitudes that are based on affect and react emotionally to empirical facts. While NFA could strengthen attachment to all types of identity attachments, including either pro-Catalan regional identity or Spanish national identity, there are two major reasons to believe that NFA could be particularly linked to support for Catalan independence.

First, for those high in NFA, emotions guide the forging of their political attitudes and preferences to the extent that individuals who are high in NFA are particularly likely "to possess extreme attitudes across a variety of issues" [48]. In the context of a region that considers secession, citizens are typically forced to support one of two following sides: a) the *status quo* or a moderate change within the system (e.g., greater decentralization); or, alternatively, b) they may support to break away with the system and secede from their host country. Following Maio and Esses, "people who like to experience emotions may be more inclined to possess extreme opinions regarding controversial issues". In this way, extreme attitudes and, potentially, behaviors, are avenues for people to experience potent emotions. In an advanced democracy such as Spain, a preference for independence—rather than accommodation—can be described as an extreme position as it involves a status-quo-challenging position on territorial politics, so I expect that individuals who are high in NFA might be more likely to support it.

A challenge to the *status quo* typically means that individuals should be willing to take or support positive action—which may range from making noise by banging pots and pans at home to violent protests in the street—to prompt or oblige the government to provide concessions to their demands. Looking at the Catalan context during the period 2012–2014, support for the Catalan self-determination movement offers individuals who favor the secession—relative to those who are indifferent or against it—numerous opportunities to engage in emotion-inducing activities. Through the influence regional politicians and media platforms in stirring up secessionist demands, the movement was characterized in the 2012–2014 period by high

levels of mass mobilization [10–14]. Pro-independence mobilization involved participating in major demonstrations, taking part in marches throughout the country, and organizing and participating in unofficial referendums, all of which are events that induce strong emotional reactions among those who join them.

The most recent Catalan history illustrates this point. Pro-secessionist Catalans mobilized more than one million people in the streets of Barcelona under the motto "We are a nation. We decide" on July 2010 to protest against the Constitutional Court's controversial decision on the Reform of the Statute of Autonomy [49]; between September 2009 and April 2011, they also held unofficial referendums on independence from Spain in which more than 800,000 citizens took part; and, as an additional example, the 2012 Catalan demonstration in Barcelona under the slogan "Catalonia, new state in Europe" gathered more than 600,000 people in the street (the official and unofficial estimates vary from 600,000 to 2 million demonstrators [50]). By contrast, those who are against the secession did not begin to organize actions against the independence until late 2017, although the frequency, scope, and intensity of these mobilizations have also been behind those organized by the secessionist movements.

During the surge of the pro-independence movement (2011–2014), individuals would perceive that supporting the secession—and not opposing it—would increase the expected likelihood of participating in emotion-inducing social and political activities. Therefore, I hypothesize that those who score high in NFA were more likely to support—rather than oppose—the Catalan independence at the time of the survey.

NFC reflects the desire to acquire a belief regardless of content. Individuals high in this motivation have a tendency to engage in and enjoy thinking, a need to structure relevant situations in meaningful ways, and make reasonable the experiential world [48, 51–54]. NFC is positively correlated with educational achievement [55], yet inconsistently connected with intelligence. Regardless, it is critical to control for education when obtaining the estimated effects.

The expected associations between NFC and support for secession are mixed. On the one hand, while individuals high in NFC might acquire more information, they might still engage in biased processing of information, be more likely to accept new arguments, new perspectives and, thus, hold more nuanced and ambivalent political preferences [44, 56, 57]. If individuals were equally exposed to all the views on a matter, one could reasonably expect that individuals who possess a high score in NFC are less likely to possess an extreme opinion.

On the other hand, the acquisition of information among people high in NFC is more elaborated and reasoned, persuasion takes place through the systematic processing rather than the heuristic processing [58]. In this way, opinions, once acquired, become less malleable to changes in the external environment. Besides, those who score high in NFC tend of have judgments that are more thoughtful, including judgments about politics [59], and have stronger attitudes making their behavior more predictable [60, 61]. In other words, people high in NFC may hold stronger, more elaborated, and resistant opinions but, at the same time, more nuanced and moderate opinions.

A long body of literature, however, shows that there is an important ideological selectivity of media use and, as a consequence, individuals are hardly ever exposed to counter-attitudinal information [62–64]. Following this argument, studies on the Catalan media consistently show an important segmentation of the media market as a function of national identity [65]. Accordingly, individuals with Catalan nationalist inclinations clearly prefer to acquire information through Catalan media, which emphasizes the arguments in favor of the secession; while non-nationalist prefer to consume state-wide, or Spanish, media, which emphasizes the arguments against the secession [66]. As a consequence, we have no clear expectations on whether those who score high in NFC would hold a stronger desire for Catalan independence.

Nevertheless, the main hypothesis—which states that those who score high in NFA should be more likely to desire the Catalan independence—should be clearest for individuals who are high in NFA and, at the same time, low on NFC. These individuals, who have been referred in prior work as "feelers" [44], should be more likely to strictly follow their emotional reaction in response to empirical facts and, as a consequence, disregarding any caveat or counter-attitudinal argument to their emotional reaction. From this argument, it follows that the expected effect of NFA on the desire for Catalan independence—an extreme view of territorial politics, an emotion-inducing preference, and a *status quo*-challenging attitude—should be exacerbated among those individuals who do not like to engage in effortful, nuanced, and complex thinking, which could provide a source of ambivalence or moderation. This is equivalent to say that those individuals who are expected to hold the least favorable view toward independence are those who combine low levels of NFA and NFC. Therefore, I expect a negative interaction effect between NFA and NFC on the support for Catalan independence.

## Methods

### Participants

37,535 participants (33,000 participants included in the final data analysis: see details of exclusion criteria below) were recruited with an average age of 48.6 (SD = 15.1) and an age range of 18–88 years. All measures were taken in one session in an online questionnaire between April 2013 and July 2013 that was available in both Catalan and Spanish. Respondents were recruited through social and mass media. Written informed consent was obtained from all participants for inclusion in the study. The recruitment process initially began as a snowball sampling in which the author posted the survey on personal social media accounts (i.e., Twitter and Facebook) and reached out faculty members from seven Catalan universities (namely, *Universitat de Barcelona*, *Universitat Pompeu Fabra*, *Universitat Autonoma de Barcelona*, *Universitat Abat Oliva*, *Universitat Rovira i Virgili*, *Universitat de Lleida*, *Universitat de Girona*) from a variety of fields. The message requested individuals to respond the survey themselves and to share it with all their contacts. In total, 3,845 faculty members were reached out and over 150 accounts shared the survey link on Twitter (advanced search query **nyu.qualtrics.com** + **lang:ca** or **lang:es** would still yield most of these accounts). Further, the survey was also publicized in the mass media, including a full-page back cover with an interview with the author and the survey link in one of the main Catalan newspapers, *El Periódico de Catalunya* [hidden link for peer-review]. The survey was administered through the Qualtrics survey platform. The survey was initiated 93,469 times. Only eligible individuals could proceed with the survey (exclusions included attempts from an IP address that had already completed the survey and IP addresses geo-located outside of Catalonia). To reduce measurement error, I filtered the data to remove careless responses: the analysis includes those who agree with the informed consent form (46,811), hold a Spanish citizenship (46,003), and provided basic background information such as sex (45,797), education (45,545), a valid age between 18 and 89 years old (40,087) and a zip code within Catalonia as a permanent residence (37,535). Additionally, the survey could not be taken twice from the same IP address, which minimizes duplicities. The survey received no funding and it was reviewed and approved by the New York University IRB Board (IRB# 13–9543).

### Survey structure

The survey was structured in four blocks. In the first block, respondents were asked about their demographic (e.g., age, gender, province of residence, educational attainment) and social background (e.g., social class, origin and mother language). In the second block, the survey

asked respondents about several psychological traits, including personality traits using the Ten-Item Personality Inventory questionnaire in Catalan or Spanish [24], items on risk acceptance, and psychological motivations (i.e., NFA and NFC). The third block of the survey asked respondents about their political values, nationalism scale, turnout, and vote intentions. Finally, the fourth block focused on the support for holding a self-determination referendum and Catalan independence. Descriptive statistics for the relevant measures can be found in the online Appendix C in S1 Appendix. The average length of the interview was 21 minutes.

## Survey representativeness

The recruitment of participants did not use a probabilistic sample. However, the survey has been weighted to approximate a representative sample of the Catalan population. Survey weights are typically inefficient in recovering representative estimates when opt-in surveys lack sufficient information due to a small sample size within each relevant demographic cell (e.g., age group, education, language). There are two major reasons that enhance the confidence in the external validity of the survey instrument. First, the massive outreach of the survey allowed us to create a sample of over 37,000 respondents. While a large sample size does not imply this to be representative, a large sample size increases the confidence in the weighting procedures because they minimize the risks associated with a small sample size within demographic cells. Second, several survey questions were carefully matched with questions from a probabilistic survey distributed in Catalonia, the Baròmetre d'Opinió Pública (BOP) of the Centre d'Estudis d'Opinió (CEO). For the purposes of linking the probabilistic to the non-probabilistic sample, I use the 2nd wave of the BOP. This has 2,000 respondents from the four Catalan provinces (Barcelona, Girona, Lleida, and Tarragona) fielded between June 3 and June 12, 2013. Using BOP survey as the base of matching, I implement a re-weighting technique for the treatment of samples when unequal selection probabilities [67]. See more information about the weighting technique in the online Appendix A in S1 Appendix. Results are robust to using the unweighted sample.

The online Appendix B in S1 Appendix compares the weighted sample frame and the general probabilistic sample from the BOP. In summary, the convenience sample significantly over-represents males (53% over 49%), supporters of Catalan independence (62% over 59%), highly-educated respondents (24% over 21%, with a college degree and above) and upper/middle and upper-class respondents (8.3% over 7.7%), and under-represents respondents of non-Catalan origin (18% over 21%), in comparison with the probabilistic sample. Note that the validity of these corrections for sample representativeness lies in the assumption that the observed respondents within each demographic cell are representative of the unobserved respondents within that demographic cell. For example, the assumption is that those in the convenience sample who reject the independence are representative of those who are outside of our sample who also reject the independence. Finally, we should also bear in mind that the validity of the weighted convenience sample to recover an unbiased estimated effect does not require a representative sample of the population but it requires that the same psychological dispositions in the convenience sample are also found in their counterparts in the mass public.

## Measures

**Desire for Catalan independence.** The desire for Catalan independence was measured with a single item: "If a referendum for independence were to be held tomorrow, what would you do?" Possible answers were "I would vote in favor of the independence", "I would vote against the independence", "I would abstain". The online Appendix C in S1 Appendix reports the descriptive statistics for all demographics and scales in the survey. This variable takes the

value of 1 if the individual answers "yes" to the question on the intention to vote in a secessionist referendum, and 0 otherwise, yet results are also robust to using this variable as categorical in the models. In total, 62% of the sample report an intention to support the secession., which is similar to the 58.5% of respondents who report an intention to vote in favor of the secession in a hypothetical referendum in the BOP 2013, wave 2.

**Need for Affect.**    The psychological need for affect was assessed using the 10-item Need for Affect Questionnaire (NAQ-S) [68]. The instrument operationalizes two factors: affect approach and affect avoidance that are typically combined in applied research [44]. The scale consists of 10 statements such as "I feel that I need to experience strong emotions regularly" or "I find strong emotions overwhelming and therefore try to avoid them". Participants respond to each statement using 7-point response scales ranging from −3 (strongly disagree) to 3 (strongly agree). As expected, the NFA items reveal a two-factor structure: the affect approach (Cronbach's alpha is 0.82) and the affect avoidance (Cronbach's alpha is 0.84). The online Appendix E in S1 Appendix reports the inter-item correlation matrix and the factor loadings. NFA scale was constructed as the aggregate sum of the affect approach and affect avoidance (reversed) items. The scale has been standardized (mean of 0, SD of 1) to ease the interpretation of coefficients.

**Need for Cognition.**    The psychological need for cognition was assessed using the 18-item Need for Cognition Questionnaire (NCQ-S) [69]. Sadowski [70] examines the validity of this scale. This scale assesses the extent to which an individual seeks and appreciates tasks requiring mental effort such as "I would prefer complex to simple problems" or "I really enjoy a task that involves coming up with new solutions to problems". The online Appendix E in S1 Appendix reports the full list of the scale items. Each item is rated on a 9-point Likert scale from 1 (extremely uncharacteristic) to 9 (extremely characteristic). The scale has been standardized (mean of 0, SD of 1) to ease the interpretation of coefficients. Factor analysis reveals that 17 out of 18 items exhibited satisfactory loadings, greater than.40, on the hypothesized latent factor while all items exhibited minor loading, less than.35, on the second and subsequent factors. Cronbach's alpha for NCQ-S in this study is.82. The online Appendix E in S1 Appendix reports the inter-item correlation matrix and the factor loadings.

**Psychological and socio-political controls.**    The empirical analyses include four sets of control variables. First, a set of personality traits measured using the Big-Five Personality Inventory (BFI), which was validated and translated into Catalan and Spanish by Renau et al. [71]. Adjusting for personality traits is important because of the existing literature documenting the association between personality factors and relevant political outcomes [53]. With regards to support for Catalan independence, earlier work shows that only the trait of Agreeableness is associated with lower support for independence [24]. Additionally, personality traits are involved in a variety of social outcomes [72–76] and political attitudes and behavior [77–79]. Given that personality traits may affect both psychological motivations and relevant social outcomes, it is important to control for them in order to estimate an effect of psychological motivations. The online Appendix F in S1 Appendix reports the descriptive statistics of the personality traits measures, correlation matrix, and validity checks. With this said, a simultaneous regression analysis reveals that the personality dimensions together explained only a small portion of the variance in the need for affect: $R2 = .07$, $F(5, 32416) = 508$, $p < .001$; and in the need for cognition: $R2 = .13$, $F(5, 32416) = 953$, $p < .001$. Therefore, they are connected yet do not overlap with the two psychological motivations studied in this paper.

Second, regression estimates include a common set of demographic and background variables: age, age squared, sex, provinces (dummies for Girona, Tarragona, and Lleida, with Barcelona as the category of reference), educational attainment, subjective social class (dummies for middle class, upper-middle-class, and upper class, with lower class as the category of

reference) and household income (in five dummies for a thousand increase to above 5, 000 euros, with less than 1, 000 euros as the category of reference).

Third, the models include two political-based controls: ideology (left-right self-placement, from 0 to 7), which might be important to explain pro-secessionist attitudes, and a dummy variable to capture whether the individual participated in the previous regional elections held in 2012. It has been suggested that pro-independence individuals might have been more politically engaged than their unionist counterparts. Therefore, accounting for the propensity to engage in politics is relevant.

Finally, the models also include controls for descent-based attributes typically regarded as salient in the Catalan case: origin (in dummies, respondent born in Catalonia with both parents born in Spain, Catalan-born with either parent Spain-born, Spain-born respondent, with Catalan-born respondents with both parents born in Catalonia as the category of reference) and mother language (in dummies, Spanish, neither and both, with Catalan as the category of reference) [13, 38]. Descent-based attributes have been found that influence support for independence and might also be associated with psychological motivations, so accounting for them is needed in the models.

## Results

This section presents the main findings of the multivariate analysis. We first focus on the main effect of NFA and NFC on the support for Catalan Independence. Then, I turn the focus on how NFC moderates the relationship between NFA and the desire for Catalan independence.

Table 1 explores the link between NFA, NFC and the desire for Catalan independence. Model 1 shows that the effect of NFA on secessionism is statistically significant at the 99%

**Table 1. The effect of Need for Affect, Need for Cognition on the desire for Catalan independence.**

| | *DV*: Desire for Catalan Independence | | | | | |
|---|---|---|---|---|---|---|
| | **(1)** | **(2)** | **(3)** | **(4)** | **(5)** | **(6)** |
| NFA | 0.24*** | 0.25*** | 0.20*** | 0.22*** | 0.22*** | 0.15* |
| | (0.07) | (0.08) | (0.08) | (0.07) | (0.07) | (0.08) |
| NFC | 0.005 | 0.08 | 0.07 | 0.001 | 0.07 | 0.07 |
| | (0.08) | (0.08) | (0.09) | (0.08) | (0.08) | (0.08) |
| NFA×NFC | | | | −0.07 | −0.09 | −0.16** |
| | | | | (0.08) | (0.08) | (0.08) |
| **Controls?** | | | | | | |
| Big 5 and SES | No | Yes | Yes | No | Yes | Yes |
| Political Factors | No | No | Yes | No | Yes | Yes |
| Descent-based Factors | No | No | Yes | No | No | Yes |
| Observations | 34,081 | 31,565 | 31,565 | 34,081 | 31,565 | 31,565 |

*Note*: Models are logistic regression models. Standard errors in parenthesis. Significance levels:

***p<0.01;

**p<0.05;

*p<0.1.

Control variables have been excluded from the regression output for the sake of space. See full models in the Online Appendix H in S1 Appendix. Multicollinearity is not detected in the models as the Variance Inflation Factors (VIF) for the explanatory variables of interest do not exceed 1.7 in any of the model specifications. More specifically, the VIF for the main-effects model in column 3 are 1.369 and 1.658 for the NFA and NFC coefficients, respectively. For the interactive model in column 6, the VIFs are 1.496, 1.635 and 1.245 for the NFA, NFC, and the interactive NFA×NFC coefficients, respectively. Given that all of these inflation factors are below the rule of thumb of 10, I understand that correlation of variables, and its interactive term, is not problematic.

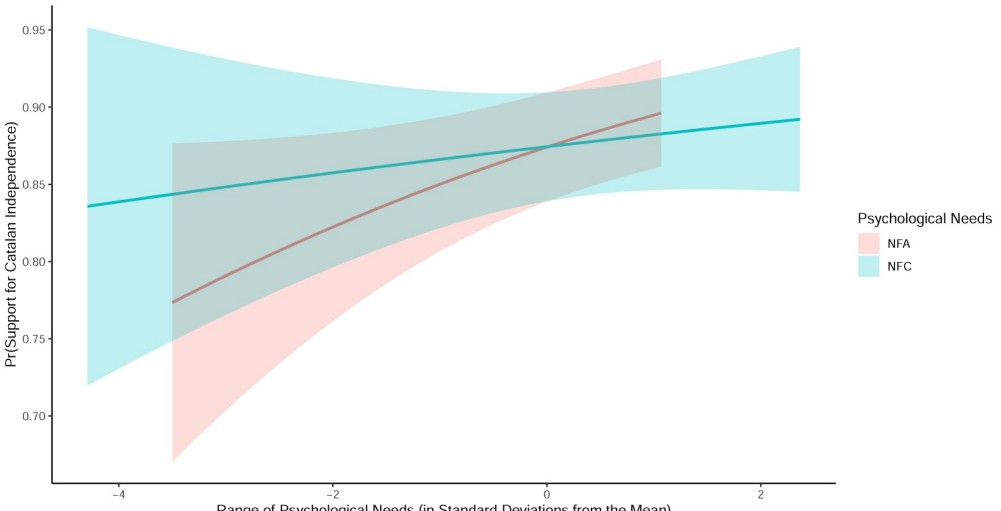

**Fig 2. Main effects of NFA and NFC on support for Catalan independence (Main-effect model).** *Note*: Predictions are based on Model 3 from Table 1.

level. This effect upholds after adjusting for respondents' personality traits and socio-economic status in model 2, as well as descent-based variables (i.e., mother language and place of origin) in model 3. Therefore, these models provide evidence that those respondents whose emotions are the main drivers of their beliefs and feel comfortable with emotionally-inducing situations are more likely to desire the Catalan independence. The same models also show that NFC has no effect on Catalan separatism.

To evaluate the magnitude of the main effects (models 1–3), Fig 2 shows the change in the predicted value of support for the independence for an individual who takes the median value across the categorical or ordinal control variables and the mean in the continuous control variables. In practice, this is equivalent to a 48-year-old man who lives in the province of Barcelona, has an income at the third level of the scale (out of the 6 income levels), holds a university degree or higher, has an average score across the Big-5 personality traits, a left-winging ideological score, turned out to vote in the most recent elections, his parents and himself are born in Catalonia, and Catalan is his mother language. In terms of probabilities, Fig 2 shows that a change in the NFA scale from -2 standard deviations below the mean to the mean increases the likelihood of voting in favor of the independence by 5.4 percent—from 81.2% to 86.6%. Similarly, an increase from the mean in NFA—when NFC is at the mean—to 1 standard deviation above the mean further rises the chance to vote in favor of the independence by an additional 2.1 percent—from 86.6% to 88.7% percent.

Fig 2 also shows that the effect of NFC on support for independence is indistinguishable from a null effect. More specifically, a shift in the NFC scale from -2 standard deviations below the mean to the mean increases the likelihood of voting in favor of the independence by 2.6 percent—from 84.8% to 87.4%, yet this increase is statistically insignificant. Similarly, an increase from the mean to 2 standard deviations negligibly increases the support for independence by 0.8 percent—from 87.4% to 88.2%. Therefore, we should conclude that there is no evidence for an average effect of NFC on support for independence.

The interaction term between NFA and NFC is negative across all model specifications. This means that the effect of NFA on support for Catalan separatism becomes particularly strong at low levels of NFC. Figs 3 and 4 illustrate this result by showing how shifts in the NFA

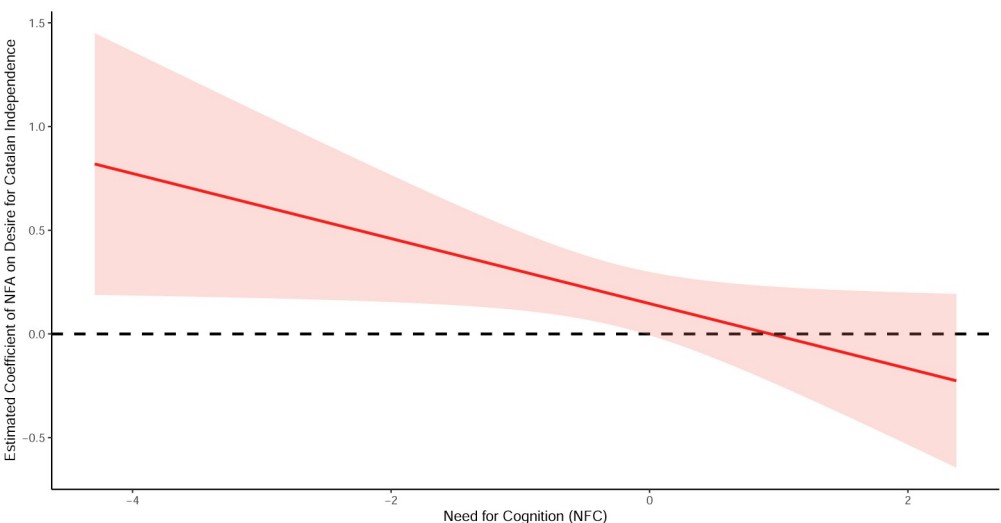

**Fig 3. Marginal effects of NFA on the desire for independence conditional on NFC (Interaction-effect models).**

and NFC score strongly influence the desire for Catalan independence. Fig 3 plots the marginal effects of NFA on the desire for Catalan independence conditional on the observed values of NFC. The effect of NFA on support for independence is positive, although it remarkably decreases along the range of values in the NFC. In fact, the effect of NFA remains significantly positive at the 95% confidence level only for those respondents whose NFC score is at the mean or below the mean of the distribution.

Similarly, Fig 4 plots the predicted values of the support for Catalan independence conditional a respondents' low or high value in NFC. In Fig 4, 2 standard deviations above the mean is considered as *high NFC* and 2 standard deviations below the mean and 2 standard deviations below the mean is considered as *low NFC*. Respondents who report a high score in NFC heavily support of Catalan independence regardless of their NFA score; their predicted value

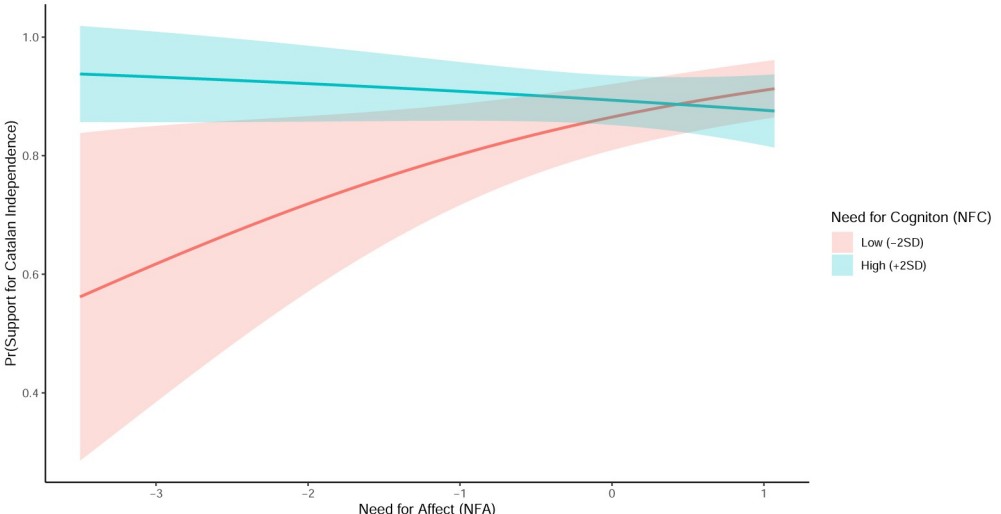

**Fig 4. Marginal effects of NFA on desire for Catalan independence conditional on NFC.** *Note*: Predictions are based on Model 6 from Table 1.

remains at a probability between 85% and 95% throughout the entire range of NFA scores. In contrast to this, those who report a low NFC score show a high chance to vote in favor of the independence only if their NFA score is high. Conditional on a low NFC score, respondents' propensity to vote in favor of the independence is about 60% at low values of NFA and about 85% at high levels of NFA. Overall, the group of respondents with the lowest support for independence are those who combine low levels of NFA and NFC at the same time.

The full regression models shown in the online Appendix H in S1 Appendix also reveal some interesting patterns. First, agreeableness was negatively associated with pro-independence support [24]. Yet, the size of the effect of NFA on support for pro-independence is about twice the effect of Agreeableness on support for Catalan independence. Second, those who support the independence tended to be ideologically more leftwing than those who support unionism. Further, those who turned out to the 2012 regional elections and, hence, more politically engaged, were more likely to support the independence. Finally, Catalan-born and those whose mother language is Catalan are also more likely to support the independence, which is consistent with extant research on the socio-political predictors of the independence [12, 13].

## Discussion and conclusion

This paper has presented an empirical investigation into the relative contributions of psychological needs for affect and cognition on the intention to vote in a potential self-determination referendum in Catalonia. By linking psychological motivations to behavior in conflict processes, this project provides a more integrative framework in the understanding of why some people want to secede from their own country while others do not. The empirical analysis uses a survey on psychological inclinations and political behavior in Catalonia, which allows analyzing the relationship between two psychological motivations that are known to shape individuals' belief formation process with pro-secessionist behavior. Building on the literature of psychological motivations [44, 48], my main expectation has been that Need for Affect (NFA) alone, and its interaction with Need for Cognition (NFC), provide the psychological foundations of attitudes toward Catalan independence.

The results reveal that those individuals who feel more comfortable with emotions, are passionate, eagerly embrace emotions, seek out for emotions, and those who believe that emotions are useful for shaping judgments and behavior are more likely to be in favor of seceding from Spain. While we find no similar direct effect for NFC on Catalan secessionism; we do find that NFA and NFC have an important negative interaction. In other words, those individuals who hold a weaker desire for Catalan independence are those who hold a low motivation for effortful reasoning (i.e., a higher tendency to process information with less elaboration, more heuristically) and, simultaneously, possess a low tendency for embracing emotions in their belief formation.

This research presents an attempt to empirically gauge the effects of psychological motivations in support of secessionist movements. Some questions remain on the table for further research. To begin with, this paper hypothesizes that stable individual differences, namely the psychological motivations of NFA and NFC, influence support for secession in a context where—as Fig 1 suggests—the desire for Catalan independence was rapidly changing. However, it remains unspecified from our empirical evidence if the psychological motivations of NFA and NFC are related to the macro-trends in the support for Catalan independence. Two behavioral models could explain our findings. On the one hand, NFA and NFC may be directly linked to the desire for Catalan independence, yet other factors such as the party in the government or economic growth—and not people's NFA and NFC—explain shifts in the overall

support for independence. If the correlation between NFA, NFC, and the desire for Catalan secession is unrelated to the changing macro-factors (e.g., party in the government, economic growth), we would observe the same association between NFA, NFC, and the desire for Catalan independence irrespective of the macro-behavioral shifts.

Alternatively, psychological motivations of NFA and NFC may not be directly associated with the outcome but may predispose people to specific attitudes and behaviors. As such, these predispositions become associated with an outcome (e.g., the desire for secession) only as a result of stimuli—a specific historical context. In fact, most political psychologists support this behavioral model when they argue about the role of personality traits in shaping political behavior. For instance, Gerber et al. [78] argue that the "Big Five traits appear to be essential aspects of individuality that shape individuals' responses to stimuli". In their discussion of the relationship between psychological traits and political ideology, scholars also refer to the concept of "elective affinities"; that is, psychological effects on political attitudes and behaviors are contingent on the stimuli, the situation, and the environment [46].

From this perspective, this paper shows a significant association between NFA and NFC and the desire for Catalan independence given the historical stimuli—i.e., emerging support for Catalan independence in the region, whether elite- and mass-based, and pro-independence monopolization of the public space. However, some questions cannot be resolved in this project. For instance, would this association exist in the early 2000s when the movement was dormant? Would the reported effects vary as a function of the strength of the movement? Future research should focus on evaluating the over-time changes in the strength of psychological factors on the support for secessionist movements using research designs better suited for this purpose (e.g., long-term longitudinal survey data). While other scholars have been able to conduct longitudinal studies on the impact of various socio-political factors on secessionism [12, 80], we unfortunately lack longitudinal data to evaluate the relationship between psychological constructs and attitudes toward secession over time.

An additional open question is the timing of the survey. The survey was distributed for a 90-day period during the summer of 2013 when the pro-independence movement had already reached its peak. Results do reflect people's tendencies at the time of the survey and they may also be a snapshot of a more dynamic process of relationships between psychological determinants and joining the secessionist movement. Theoretically, it is possible that secessionist movements share characteristics of threshold models of collective behavior [81, 82] by which the tendency to join movements may be explained by different factors at different points in time.

In that vein, people's pro-secessionist behavior could be the product of an external shock moving the number of supporters above a tipping point. This would boost the movement to grow larger in a self-reinforcement process until reaching a natural ceiling point. Under this theoretical framework, results from a survey that divides citizens between pro-independence and unionist ought to be interpreted as reflecting the psychological differences between those above and below the threshold in a specific time. However, a shift in the perceived costs and benefits would lead to a shift in the acceptance threshold of a secession. And, if dispositional traits determine this acceptance threshold, they would also determine changes in the intention to vote for the secession. Thus, capturing a dynamic relationship between motivations and a relative threshold to join a secessionist movement presents a great potential for future research. However, it is still far from feasible given the current availability of data and clearly beyond the scope of this project.

This paper is agnostic on what specifically allowed for the connection between psychological NFA and secessionism in the early stages of the movement in 2013. Some scholars would argue that regional elites played a role in constructing an emotionally-driven environment to

push for independence, which led those who were more passionate and more eagerly embraced emotions to join the movement. This is consistent with this paper's findings and, certainly, this process might have come at the same time as many Catalans changed their dual Spanish-Catalan identities to hold an exclusive Catalan identity [13, 80]. Furthermore, some scholars have suggested that the monopolization of the public space by secessionist symbols have created a context of persuasion through two major channels: local media and continuous social pressure [66]. While it is plausible that a change in identity and the activation of psychological needs among the population could have arisen at the same time as the regional elites the began pushing for the independence through local media and social pressure, I acknowledge that I lack the longitudinal data necessary to link these macro-processes, attitudinal changes, and shifts in psychological needs in time.

Another aspect that requires further investigation has to do with the generalizability of these findings to other secessionist movements. The Catalan case has been often regarded as a prototypical case of a self-determination movement in a developed country—i.e., a peripheral region that is relatively advantageous in economic terms and a language/cultural singularity within the country. Given that NFA and NFC are quite consistent psychological constructs, the evidence presented here might generalize to most similar cases such as Flanders, Quebec, the Basque Country, and Scotland. Following the theoretical framework, I could speculate that the connection between NFA and NFC on secessionism is likely to require a context similar to that lived in Catalonia in 2013, which involved the monopolization of the public space by secessionist symbols and the presence of large-scale pro-independence mobilizations. Further research will also be required to examine whether contexts such as the Brexit movement could have shown a similar relationship between NFA and NFC and Brexit support. Outside of the developed world, however, many secessionist movements are strongly influenced by the dynamics of armed conflict. Hence, further research should assess whether macro-level processes (e.g., armed conflict) could wash away the effect of psychological factors on support for secessionist movements.

Additionally, this project relies on self-reports of psychological aspects, attitudes, and intentions of behavior. Therefore, the survey variables might capture a mix of accurate psychological, attitudinal, and behavioral tendencies as well as measurement error and bias that comes from the survey instrument itself. However, we should note that the presence of measurement error, if anything, would attenuate the relationship among variables, which means that the reported effects may under-estimate the true effects.

Altogether, this article contributes to the impact of psychological traits on support for independence with an original dataset. An important contribution of this article is to present evidence to support the view that psychological motivations impact individuals' propensity to support secessionist movements. This view is consistent with the claim that psychological variables such as personality, psychological motivations, and risk attitudes can be viewed as factors that predate social and political influences, rather than being caused by them [77, 79]. From this perspective, as it occurs with the role of personality traits, psychological motivations can be conceptualized as distant factors that shape, independently or in conjunction with other environmental variables, the proximal political attitudes upon which researchers tend to focus on.

## Supporting information

**S1 Appendix.**
(PDF)

## Author Contributions

**Conceptualization:** Joan Barceló.

**Data curation:** Joan Barceló.

**Formal analysis:** Joan Barceló.

**Investigation:** Joan Barceló.

**Methodology:** Joan Barceló.

**Project administration:** Joan Barceló.

**Visualization:** Joan Barceló.

**Writing – original draft:** Joan Barceló.

**Writing – review & editing:** Joan Barceló.

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
