## [Decision Letter · Decision Letter 0]

15 Feb 2022

PONE-D-21-32282Need for Affect, Need for Cognition, and the Desire for IndependencePLOS ONE

Dear Dr. BARCELO SOLER,

Thank you for submitting your manuscript to PLOS ONE. After careful consideration, we feel that it has merit but does not fully meet PLOS ONE’s publication criteria as it currently stands. Therefore, we invite you to submit a revised version of the manuscript that addresses the points raised during the review process. Please consider reviewers' comments carefully and revise accordingly. I summarize major points raised by the reviewers.

1. The research needs to situate itself in the previous literature on the support of political independence in Catalonia or elsewhere (e.g., reviewer 1's points 2, 4, & 7; reviewer 2' point in linking NFA and NFC to the identity and instrumental models).

2. The author(s) need to more carefully lay out what has been controlled for (and how the controls were linked to the previous literature).

3. More detailed information should be given with regards to the development of the scales (reviewer 1' point 5, reviewer 2's comments regarding results that are not the foci of the authors' hypotheses).

4. I also notice that the two types of needs were highly correlated (r = .41) and feel that the interaction term is going to be problematic (how can you interpret something that is overlapping in nature that has interaction?) Also, judging from the interaction terms in the models, it is only in model 6 that this interaction becomes significant.

5. I am also not quite sure that I understand the logic of your proposed hypotheses. If NFA is a "general motivation to approach or avoid emotions or emotional inducing situations and includes the beliefs that emotions are useful for shaping judgments and behavior"; how does this general motivation translate to the positive attitudes toward independence? I am thinking that interaction terms of NFA and some demographic characteristics that may reflect the identity approach would be more coherent in the current framework. The same set of questions could be applied to the hypotheses of NFC. It seems to me that NFA and NFC catch different ways of processing information. However, how these processes could be linked to positive attitudes toward independence are not clearly laid out.

We look forward to receiving your revised manuscript.

Kind regards,

I-Ching Lee

Academic Editor

PLOS ONE

Journal Requirements:

a) Did participants provide their written or verbal informed consent to participate in this study?

5. Please ensure that you include a title page within your main document. You should list all authors and all affiliations as per our author instructions and clearly indicate the corresponding author.

Reviewers' comments:

Reviewer's Responses to Questions

**Comments to the Author**

1. Is the manuscript technically sound, and do the data support the conclusions?

Reviewer #1: Partly

Reviewer #2: Partly

2. Has the statistical analysis been performed appropriately and rigorously? 

Reviewer #1: Yes

Reviewer #2: Yes

3. Have the authors made all data underlying the findings in their manuscript fully available?

Reviewer #1: Yes

Reviewer #2: No

4. Is the manuscript presented in an intelligible fashion and written in standard English?

Reviewer #1: Yes

Reviewer #2: Yes

5. Review Comments to the Author

Reviewer #1: This study provides results suggesting that some particular psychological attributes (“need for affect” and “need for cognition”) might contribute to the expressed support for political secession on a Western European democracy. It does so by studying the relationships between these individual psychological traits and the willing to secede on a huge sample (N>33.000) of Catalonian citizens who answered an online survey, during several months at 2013, at the peak of the secessionist effervescence that erupted in that Northeastern Spanish region at that time.

The study is well written, wisely argued and the technical analysis of the findings appears professional and sound. The justification for introducing psychological vectors as possible additional mediators of political decisions in turmoil situations is substantiated using the available literature on the issue. The discussion of the consistency and implications of the main findings, as well as some of its limitations, is also cogently presented.

The findings deserve to be known because they suggest that behind an unexpected urge for secession, in addition to identity, economic or other political vectors, individual psychological traits related mainly to emotional appetites played a non trivial role on a particular moment, at least, of the lively and enthusiast Catalonian secessionist movement (during several years, till it did’nt success failing to attain its goals).

The paper needs, however, substantial revision and refinements on different parts before being considered acceptable for publication.

Those are the main points that have to be addressed:

1. At the introduction and through the whole paper the desire for political independence in Catalonia is described as if it was an spontaneous and grass-rooted movement. It obviates that it had a continuous and firm direction by a Regional Government and an Autonomous Parliament both dominated by secessionist parties. As an example, Figure 1 is used to ilustrate the abrupt increase of the willing to secede between 2010-2013, neglecting to specify that such abrupt increase was, in fact, at 2012-2013, around the regional elections of 2012 that were called by a nationalist-lead Government that had changed from a decades-long moderate politics centered on enhancing devolution from Spain to a direct secession pathway.

2. In this respect, the paper fully obviates a rich literature that has showed the relevant role played by regional elites and particularly the demanding outbid between regional nationalist parties on igniting and feeding the movement (Barrio and RdzTeruel 2017; Barrio and Field, 2018; Rdz. Teruel and Barrio 2021); or the crucial propagandistic role played by local media under direct or indirect influence of a highly resourceful Regional Government (Oller et al, 2019).

3. The huge online sample is a convenient one but it’s not representative of Catalonian population. This is duly recognized but has to be addressed in a much more consistent way. Despite the cautions taken by weighing and adjusting scores to those of a corresponding CEO profile in one of its regular Barometers surveys, the final profile of the sample is biased and consists on a peculiar cluster of Catalonian citizenry: educated, middle and upper class, mature age males, pro-leftist, pro-secession individuals born inside Catalonia clearly predominated. So, the results might illustrate tendencies and propensities of an important fraction of Catalonian citizenry but not those from the majoritarian population at all. In addition to the peculiar profile provided by these socioeconomic variables, a single measure illustrates that: 62% in favor of secession is an oddity, both in the long series of CEO and CIS surveys, and in all electoral results in the last decade. So the relevance of that should be discussed much more thoroughly.

4. The author avoids contrasting the potencies of his main psychological variables here with other personality vectors previously studied by him in another study (Barceló 2017), on the same sample. That’s particularly surprising because his main predictor there “Agreableness”, is highly charged by emotional tones, and emotion appetites play a crucial role on his interpretation of the findings. In addition, the absence of a proper discussion of the role played by “Need for cognition” and its plausible relation to “Openess”, Intelligence or Education levels is also disappointing, because it reduces the analyses of individual psychological factors to a very narrow and perhaps inconsistent display. So, a full treatment and discussion of these complementary measures seems peremptory and might perhaps add to the role proposed for individual psychological factors on political options.

5. No references are given about translation and validation of the scales used to measure “need for affect” and “need for cognition”. The Catalan and Spanish translations (Supp. Information) are very poor, with obvious ortographical and gramatical errors, and that should not be obviated when discussing limitations and weaknesses of the findings.

6. References to previous studies that have explored differential emotional landscapes characterizing the secessionist and unionist segments of Catalonian citizenry seems mandatory at the discussion (Oller et al 2018, Tobeña 2016, 2017).

7. There is also an insufficient reference to studies that have showed important variations both on national identities and socioeconomic measures through the whole period of Catalonian secessionist effervescence, in longitudinal studies (i.e. Oller et al, 2019; 2020; 2021). The diagnoses of a transversal study made on a very particular moment should be compared and complemented with systematic and already well known longitudinal landscapes.

8. Regarding that, the lack of contrast of the obtained results with results from another similar online massive survey taken, at a different and crucial time during the secession bid, is a major weakness insufficiently discussed.

Only if all these insufficiencies and caveats are adequately solved the present paper should go to the press. The findings, though transient and only suggestive, are clearly interesting and they deserve to be known.

Reviewer #2: A review of the manuscript “Need for Affect, Need for Cognition, and the Desire for Independence”.

This study sets out to examine the role of two psychological factors in explaining intention to vote for Catalan independence from Spain. The study has an impressive sample size of over 35,000 Catalan respondents who are asked a battery of relevant questions, including two scales with good psychometric properties.

My general assessment of the paper is that it has a lot of potential but needs to be developed further, both theoretically and statistically. The author discusses two main reasons that have been suggested in the literature for why people support independence movements, the identity model, and the instrumental model. Why then, did the author choose to measure NFA and NFC in this study? There is no attempt to link NFA and NFC to the identity and instrumental models. In addition, a deeper dive into previous research relevant to the potential role of NFA and NFC in support for independence movements would have strengthened the paper.

The study includes the ten-item personality inventory without explaining why or what the results mean. Same can be said for some of the other demographic and attitudinal variables.

As for the reporting of data I would have liked to see descriptive statistics (not just correlations in the appendix) and reporting of explained variance in the regression tables.

The more detailed regression table in the appendix also reveals several curiosities I suggest the author incorporates formally into the paper. The role of left-wing ideology, higher education and income, turnout in 2012 (what does that stand for?), and the clear importance of speaking Catalan.

I was quite puzzled by the high predicted support from the models. While 62% of the sample supported independence for almost all the predictions mentioned in the manuscript they are much higher than that (usually around 85%). Maybe there is a clear statistical explanation for this, but I think it has to be provided.

Finally, I think it may be more appropriate to frame the main finding in terms of the only group that shows the weakest support for independence. The only group in the four-cell matrix of high-low NFC and high-low NFA is the “low-NFC and low-NFA” group.

6. PLOS authors have the option to publish the peer review history of their article (what does this mean?). If published, this will include your full peer review and any attached files.

Reviewer #1: No

Reviewer #2: No

---

## [Author Response · Author response to Decision Letter 0]

27 Apr 2022

Dear Academic Editor, I-Ching Lee,

Thank you for the opportunity to resubmit this manuscript. I have attached the response to reviewers as a separate file in the "Attach Files."

Sincerely, 

The Author

---

## [Decision Letter · Decision Letter 1]

29 Jul 2022

PONE-D-21-32282R1Need for Affect, Need for Cognition, and the Desire for IndependencePLOS ONE

Dear Dr. BARCELO SOLER,

Thank you for submitting your manuscript to PLOS ONE. After careful consideration, we feel that it has merit but does not fully meet PLOS ONE’s publication criteria as it currently stands. Therefore, we invite you to submit a revised version of the manuscript that addresses the points raised during the review process. The current manuscript, though, has improved, according to reviewer 1, is not evaluated favorably by reviewer 3. I read the manuscript myself and agreed with the major concerns raised by reviewer 3. Please explain carefully how psychology needs could be linked with support for the independence and let this explanation be the focus of your introduction.

It seems to me that the proposed associations of the need for affect and need for cognition do not pertain only to the case in Catalonia and if you could find additional data to support/replicate the current findings, it will greatly increase the merits of the current manuscript.

We look forward to receiving your revised manuscript.

Kind regards,

I-Ching Lee

Academic Editor

PLOS ONE

Reviewers' comments:

Reviewer's Responses to Questions

**Comments to the Author**

1. If the authors have adequately addressed your comments raised in a previous round of review and you feel that this manuscript is now acceptable for publication, you may indicate that here to bypass the “Comments to the Author” section, enter your conflict of interest statement in the “Confidential to Editor” section, and submit your "Accept" recommendation.

Reviewer #1: All comments have been addressed

Reviewer #3: (No Response)

2. Is the manuscript technically sound, and do the data support the conclusions?

Reviewer #1: Yes

Reviewer #3: No

3. Has the statistical analysis been performed appropriately and rigorously? 

Reviewer #1: Yes

Reviewer #3: Yes

4. Have the authors made all data underlying the findings in their manuscript fully available?

Reviewer #1: Yes

Reviewer #3: Yes

5. Is the manuscript presented in an intelligible fashion and written in standard English?

Reviewer #1: Yes

Reviewer #3: Yes

6. Review Comments to the Author

Reviewer #1: All comments and suggestions have been thoroughly ans satisfactorily addressed. The paper has greatly improved

and must go swiftly to the print.

My only suggestion is for the Abstract: It should refer to the absence of relation between NFC and support for independence .

Some minor errors remain in the Catalan-language versions of questionnaires at the Appendix: i.e. "complexos" (instead of "complexes"; "requereixi" (instead of "requerixi"); Catalan sentences do not demand a proposition particle before direct complements.; the translation looks rather "formal" and "bureocratic", distanced from common use of the language. But that cannot be changed.

Reviewer #3: This is an interesting work connecting psychological underpinnings (of NfA and NfC) with the Catalonian independence support. The author uses a large n, non-representative adult sample to test the association between psychological motives and support for the secessionist movement in Catalonia. As much as one could find a number of interesting aspects in this manuscript, the theoretical disconnect makes it hard to comprehend the real research goal and the potential inferences in this study. I thus cannot be supportive of its publication in its current condition. I further explain my comments below.

First, the primary gap in this work is the disconnect between the psychological drives (i.e., NfA and NfC) that tap individual differences in the sample and preferences on the Catalonian secession. Generally, work that uses NfC or NfA relate these factors to thought processes (as used in the original seminal studies) or to mid-level concepts (e.g., Big5, different types of thinking processes, compassion/empathy). These concepts cannot be treated as factors that one just wants to control for in a study. There has to be some kind of theoretical reasoning linking how NfC and NfA explain actual preferences on secession. That is the biggest aspect missing in the current manuscript.

Second, Cognition and Affect are two of the basic constructs of thought processes that capture how and why people think/evaluate/prefer the way they do. These constructs explain the mechanism of whether a person’s thinking process is driven by cognitive assessments or affective evaluations. Similar concepts such as the Need to Evaluate or Uncertainty Avoidance explain the mechanism of thinking process, but not the behavioral outcome such as vote choice (or support for independence). The author repeatedly tries to connect emotions and cognition with the behavioral outcome in the findings section, but the evidence and discussions are quite weak.

Since I don’t think that there is a decent theoretical foundation associating NfC and NFA with support for independence, I may recommend the following alternatives: One option to link NfC and NfA to the support for independence may go through the Big5. One of the Big5 dimensions may promote higher/lower support for the independence. There is preceding work on how Agreeableness, Conscientiousness, or other factors may relate to ideologies, voting behavior, strategic decisions, and freedoms. Another alternative could be through the level of political learning and political knowledge. Both of these factors relate to thinking processes that NfC and NfA are associated with. Politically knowledge could reveal some interesting results.

On additional issue with this manuscript is the generalizability of the results. Can we relate the findings to other secessionist movements across the globe? Given that NfC and NfA are quite consistent psychological constructs, would they explain similar preferences in Canada, Scotland, or China? Or, could we say anything about Brexit? The authors touches upon that but could not develop the idea as the theoretical foundations are quite weak. If supported properly, I think that would be a quite important inference.

7. PLOS authors have the option to publish the peer review history of their article (what does this mean?). If published, this will include your full peer review and any attached files.

Reviewer #1: No

Reviewer #3: No

---

## [Author Response · Author response to Decision Letter 1]

11 Oct 2022

The Editor

Thank you for submitting your manuscript to PLOS ONE. After careful consideration, we feel that it has merit but does not fully meet PLOS ONE’s publication criteria as it currently stands. Therefore, we invite you to submit a revised version of the manuscript that addresses the points raised during the review process.

The current manuscript, though, has improved, according to reviewer 1, is not evaluated favorably by reviewer 3. I read the manuscript myself and agreed with the major concerns raised by reviewer 3. Please explain carefully how psychology needs could be linked with support for the independence and let this explanation be the focus of your introduction.

• Thanks for your guidance on these points. I have added a more developed theoretical explanation in the theory section, as well as in the introduction. To be clear, the theoretical framework proposed here is that NFA and NFC are connected to the outcome of interest, support for the independence, conditional on the Catalan movement in 2013. In 2013, the movement was characterized by two major factors: the monopolization of the public space by secessionist symbols and the presence of large-scale pro-independence mobilizations. These conditions constitute the scope conditions of the paper and, as I argue, allowed NFA and NFC to be associated with pro-independence support. I hope the reviewer and the editor appreciate these clarifications in the paper.

It seems to me that the proposed associations of the need for affect and need for cognition do not pertain only to the case in Catalonia and if you could find additional data to support/replicate the current findings, it will greatly increase the merits of the current manuscript.

• I agree that the associations that are reported in this paper could travel well to other papers. In a way, the argument made in the paper is not a Catalan-specific argument but a general argument about behavioral tendencies in situation of an emergence of secessionist or transgressing movements. In my view, the generalizibility of the argument and, therefore, the expected generalizibility of the findings is a merit (and not a demerit) of the manuscript. With this said, I am not aware of any survey data that has included the NFA and the NFC in a similar context that could be used to test or replicate the same argument in a different context. If the editor or any of the reviewers are aware of this, I would be happy to add a new section. Otherwise, I have to leave this point for further research. The lack of comparable data highlights the novelty of the data analyzed in this paper. It is the first time, to the best of my knowledge, that the NFA and NFC scales are embedded in a political survey in the context of an emerging secessionist or transgressing movement. The advantage of this is that, for the first time, the associations between NFA and NFC and the support for these movements can be examined, yet it comes at the cost of not being able to find comparative data to empirically define the scope conditions of such associations. Overall, I hope that the editor and the reviewer appreciate the contribution of this manuscript to the literature.

Reviewer 1

(1) All comments and suggestions have been thoroughly ans satisfactorily addressed. The paper has greatly improved and must go swiftly to the print.

• Thanks for your support.

(2) My only suggestion is for the Abstract: It should refer to the absence of relation between NFC and support for independence .

Some minor errors remain in the Catalan-language versions of questionnaires at the Appendix: i.e. ”complexos” (instead of ”complexes”; ”requereixi” (instead of ”requerixi”); Catalan sentences do not demand a proposition particle before direct complements.; the translation looks rather ”formal” and ”bureocratic”, distanced from common use of the language. But that cannot be changed.

• Thanks for your guidance throughout the revision process. I have implemented these changes.

Reviewer 3

(1) This is an interesting work connecting psychological underpinnings (of NfA and NfC) with the Catalonian independence support. The author uses a large n, nonrepresentative adult sample to test the association between psychological motives and support for the secessionist movement in Catalonia. As much as one could find a number of interesting aspects in this manuscript, the theoretical disconnect makes it hard to comprehend the real research goal and the potential inferences in this study. I thus cannot be supportive of its publication in its current condition. I further explain my comments below.

First, the primary gap in this work is the disconnect between the psychological drives (i.e., NfA and NfC) that tap individual differences in the sample and preferences on the Catalonian secession. Generally, work that uses NfC or NfA relate these factors to thought processes (as used in the original seminal studies) or to mid-level concepts (e.g., Big5, different types of thinking processes, compassion/empathy). These concepts cannot be treated as factors that one just wants to control for in a study. There has to be some kind of theoretical reasoning linking how NfC and NfA explain actual preferences on secession. That is the biggest aspect missing in the current manuscript.

Second, Cognition and Affect are two of the basic constructs of thought processes that capture how and why people think/evaluate/prefer the way they do. These constructs explain the mechanism of whether a person’s thinking process is driven by cognitive assessments or affective evaluations. Similar concepts such as the Need to Evaluate or Uncertainty Avoidance explain the mechanism of thinking process, but not the behavioral outcome such as vote choice (or support for independence). The author repeatedly tries to connect emotions and cognition with the behavioral outcome in the findings section, but the evidence and discussions are quite weak.

Since I don’t think that there is a decent theoretical foundation associating NfC and NFA with support for independence, I may recommend the following alternatives: One option to link NfC and NfA to the support for independence may go through the Big5. One of the Big5 dimensions may promote higher/lower support for the independence. There is preceding work on how Agreeableness, Conscientiousness, or other factors may relate to ideologies, voting behavior, strategic decisions, and freedoms. Another alternative could be through the level of political learning and political knowledge. Both of these factors relate to thinking processes that NfC and

NfA are associated with. Politically knowledge could reveal some interesting results.

• I appreciate the thoughtful comments and suggestions of the reviewer. The reviewer is raising important concerns on whether psychological needs such as NfA and NfC may conceptually have a direct effect on preferences and behavior.

The theoretical reasoning of this manuscript considers that Nfa and NfC can be predictors of preferences and behaviors conditional on individuals’ stimuli, experiences, or the context. In other words, in a given context, how you process information (cognitive assessments or affective evaluations) affects what you think about an issue and shapes how you behave.

Imagine a scenario where people watches an emotional advertisement that seeks to persuade them to donate money for an NGO. All else equal, we could reasonably expect that those high in NfA would be more likely to donate after watching the ad that those who are low in NfA. In other words, given a given context, NfA can co-vary with preferences and behavior. As the reviewer points out, this is an assumption made in the manuscript.

Translating this to the content of this manuscript, pages 8-12 elaborate on how the Catalan context makes individuals’ psychological drives to co-vary with preferences toward secessionism.

”During the surge of the pro-independence movement (2011-2014), individuals would perceive that supporting the secession—and not opposing it—would increase the expected likelihood of participating in emotion-inducing social and political activities.” I note that ”pro-independence mobilization involved participating in major demonstrations, taking part in marches throughout the country, and organizing and participating in unofficial referendums, all of which are events that induce strong emotional reactions among those who join them.” ”Therefore, I hypothesize that those who score high in NFA were more likely to support—rather than oppose—the Catalan independence at the time of the survey.” (p.10).

The arguments provided in the manuscript emphasize the political context as a stimuli that would call for people high in NFA but not those low in NFA to support the independence. Obviously, if individuals would perceive that supporting the union would increase the expected likelihood of participating in emotion-inducing social and political activities, we would then expect NFA to be associated with the support for the union. I, hence, acknowledge that the hypothesis are derived from a general theory of behavior but specifically applied to the context of the Catalan conflict as it was in 2013.

This manuscript is not the first to explore the direct effects of NFA and/or NFC on preferences and behaviors. Most obviously, the seminal work by Maio and Esses (2001) already considers a direct effect of NFA on preferences. They state that “people who like to experience emotions may be more inclined to possess extreme opinions regarding controversial issues” (538). In other words, given a controversial issue, Maio and Esses (2001)’s prediction is that those high in NFA would hold the extreme preference. This is a statement predicting a direct connection between NFA and a preference.

Furthermore, extant research documents direct associations between NFA and numerous attitudinal and behavioral outcomes such as:

– Lins de Holanda Coelho et al. (2018) examine the effect of affective traits, including NFA, on the use of different types of drugs.

– Rosenbaum and Johnson (2016) study the role of psychological drivers on the enjoyment of unspoiled stories. The authors find that those low on need for cognition hold a selective preference for spoiled stories, whereas individuals with a high need for affect enjoyed unspoiled stories more.

– Cramer et al. (2021) study how NFA and NFC may be clinically relevant for persons at escalated risk for suicide.

– Yildizeli Topcu (2021) look at the role of NFA on nurse’s care-giving approach.

– Kang (2020) investigates the effects of NFA on attitudes toward online games.

Likewise the numerous studies on the direct role of NFA on attitudes and behaviors, the role of NFC has been similarly examined across a wide number of disciplines.

Regarding the direct role of NFC on attitudes and behaviors, Zerna, Strobel and Strobel (2021) is illustrative as they conducted a systematic review covering over 100 studies analyzing the direct effect of NFC on well-being. The outcomes in the studies, which are summarized in this review, include all kinds of preferences and behavioral intentions such as religiosity, college dropout, study performance, vaccine uptake, conspiracy beliefs, smoking, self-reported health, alcohol consumption, materialism, sleep deprivation, life satisfaction, decisional procrastination, study satisfaction, psychological wellbeing, cancer recurrence worry, academic engagement, self-efficacy, among many others. The meta-analysis is available at: https://psyarxiv.com/p6gwh/

Overall, this manuscript goes together with extant literature treating the NFA and NFC as factors – or underlying psychological motives - that shape how people process information in a particular context, which affects what people think about an issue and how they behave. I have revised the manuscript by adding a footnote (ft. 4, pg. 8) that makes explicit that the directional hypotheses are contingent to the context of the emergence of the Catalan secessionist movement, and referred to the extant literature that makes a direct connection between these psychological motivates and attitudes and behaviors.

(2) On additional issue with this manuscript is the generalizability of the results. Can we relate the findings to other secessionist movements across the globe? Given that NfC and NfA are quite consistent psychological constructs, would they explain similar preferences in Canada, Scotland, or China? Or, could we say anything about Brexit? The authors touches upon that but could not develop the idea as the theoretical foundations are quite weak. If supported properly, I think that would be a quite important inference.

• I appreciate the thoughtful suggestion of the reviewer. I have expanded the paragraph on p. 26 to capture this idea. However, I have to keep this in the realm of speculation as further research will be required to evaluate the extent to which the findings from Catalonia will generalize to other secessionist movements or to other exclusivity movements such as the Brexit. Given the scope conditions of the Catalan context described in p. 9-10, namely the role of the regional elites in stirring up secessionist demands and the monopolization of the public space through emotion-inducing activities by the secessionist movement (protest, demonstrations), I can only speculate that it seems sensible that these contextual conditions are required to allow for individuals high in NFA, and also those high in NFA and low in NFC, to b willing to support secessionist or transgressing movements.

See the paragraph on p. 26: “Another aspect that requires further investigation has to do with the generalizability of these findings to other secessionist movements. The Catalan case has been often regarded as a prototypical case of a self-determination movement in a developed country—i.e., a peripheral region that is relatively advantageous in economic terms and a language/cultural singularity within the country. Given that NFA and NFC are quite consistent psychological constructs, the evidence presented here might generalize to most similar cases such as Flanders, Quebec, the Basque Country, and Scotland. Following the theoretical framework, I could speculate that the connection between NFA and NFC on secessionism is likely to require a context similar to that lived in Catalonia in 2013, which involved the monopolization of the public space by secessionist symbols and the presence of large-scale pro-independence mobilizations. Further research will also be required to examine whether contexts such as the Brexit movement could have shown a similar relationship between NFA and NFC and Brexit support. Outside of the developed world, however, many secessionist movements outside the developed world are strongly influenced by the dynamics of armed conflict. Hence, further research should assess whether macro-level processes (e.g., armed conflict) could wash away the effect of psychological factors on support for secessionist movements.”

---

## [Decision Letter · Decision Letter 2]

12 Dec 2022

PONE-D-21-32282R2Need for Affect, Need for Cognition, and the Desire for IndependencePLOS ONE

Dear Dr. BARCELO SOLER,

Thank you for submitting your manuscript to PLOS ONE. After careful consideration, we feel that it has merit but does not fully meet PLOS ONE’s publication criteria as it currently stands. Therefore, we invite you to submit a revised version of the manuscript that addresses the points raised during the review process.

 I have secured two reviewers' evaluations of the current manuscript and while one endorses the publication of the manuscript, one is more skeptical. The reviewer suggests a stronger link of NFC and NFA with political attitudes or judgements and recommends some papers that may be relevant for citations. Please consider incorporate them in the current manuscript. My reading is that it is very close to be accepted for publication but some revisions are needed. In addition to the literature the reviewer suggested, changes to some wording may be incorporated. On page 1, you refer NFC and NFA as traits but motivations throughout the manuscript, please replace traits with motivations.  Because you have a survey data set which was collected one time, please be very careful when you use the causal terms. In addition, in the sample section, you started with 37,535 participants being recruited, but this is not true. Information provided elsewhere suggested that you recruited at least 47,148 individuals or 93,469 (survey link being clicked). Please revise to remove the inconsistent information and making sure that it is clear that the number of 37,535 participants were the ones that are further analyzed. 

We look forward to receiving your revised manuscript.

Kind regards,

I-Ching Lee

Academic Editor

PLOS ONE

Journal Requirements:

Reviewers' comments:

Reviewer's Responses to Questions

**Comments to the Author**

1. If the authors have adequately addressed your comments raised in a previous round of review and you feel that this manuscript is now acceptable for publication, you may indicate that here to bypass the “Comments to the Author” section, enter your conflict of interest statement in the “Confidential to Editor” section, and submit your "Accept" recommendation.

Reviewer #3: (No Response)

Reviewer #4: (No Response)

2. Is the manuscript technically sound, and do the data support the conclusions?

Reviewer #3: Partly

Reviewer #4: Yes

3. Has the statistical analysis been performed appropriately and rigorously? 

Reviewer #3: Yes

Reviewer #4: Yes

4. Have the authors made all data underlying the findings in their manuscript fully available?

Reviewer #3: Yes

Reviewer #4: Yes

5. Is the manuscript presented in an intelligible fashion and written in standard English?

Reviewer #3: Yes

Reviewer #4: Yes

6. Review Comments to the Author

Reviewer #3: I enjoyed reading the revised version of the ms. Although we see minor changes in the written text, the response letter gives a bit more context and detail to reviewer comments. I must admit that I am still skeptical about the strength of the theoretical connection between NFA&NFC and the behavioral support for secessionism. Two important concerns come to mind:

First, the author claims that we may not have a preference about what those who score high in NFC would desire for Catalan independence. I disagree because one of the principal aspects of NFC is that people high in need for cognition tend to have persistent attitudes that are resistant to counterarguments (Haugtvedt and Petty, 1992). Besides, those who score high in NFC tend of have judgments that are more thoughtful, and have stronger attitudes making their behavior more predictable (Petty et al., 1995). With that, why could not we have clear predictions about high NFC people’s preferences for independence? The theory could lead us make expectations prior to empirical insignificant effects.

Second, I think the author needs to be selective in the work associating NFA&NFC and the behavioral outcomes. Going back to the original works of Cacioppo and Petty in 1980s up until today, there must have been more than a few hundred publications using these two psych domains across an array of attitudinal and behavioral outcomes. Yet, the author lists (in the response letter) some that vaguely relate to political attitudes or behavior. I thus recommend referring to recently published work on these and related psych domains in the field with relevance to political attitudes and judgments.

One minor issue: On page 26, in the revised section, “outside of the developed word” is repeated.

Cited works

Haugtvedt, Curtis P and Richard E Petty (1992) Personality and Persuasion: Need for cognition moderates the persistence and resistance of attitude changes. Journal of Personality and Social Psychology 63(2): 308–319.

Petty, Richard E, Curtis Haugtvedt and Stephen M Smith (1995) Elaboration as a Determinant of Attitude Strength: Creating attitudes that are persistent, resistant, and predictive of behavior. In Richard E Petty and John A Krosnick (eds) Attitude Strength: Antecedents and Consequences. Mahwah, NJ: Lawrence Erlbaum Associates, 93–130.

Reviewer #4: This is an interesting piece that builds on research that incorporates NFA and NFC into political decision making. The large-n survey is also very impressive and allows the authors to estimate the two-way interaction of continuous variables well.

7. PLOS authors have the option to publish the peer review history of their article (what does this mean?). If published, this will include your full peer review and any attached files.

Reviewer #3: No

Reviewer #4: No

---

## [Author Response · Author response to Decision Letter 2]

16 Dec 2022

Thank you for the opportunity to resubmit this manuscript. I have revised the manuscript, taking seriously the last few points raised by the editor and reviewer #3. My comments and responses are shown below each point in the response memo.

---

## [Editor Report · Decision Letter 3]

3 Jan 2023

Need for Affect, Need for Cognition, and the Desire for Independence

PONE-D-21-32282R3

Dear Dr. BARCELO SOLER,

We’re pleased to inform you that your manuscript has been judged scientifically suitable for publication and will be formally accepted for publication once it meets all outstanding technical requirements.

Kind regards,

I-Ching Lee

Academic Editor

PLOS ONE
---

## [Editor Report · Acceptance letter]

30 Jan 2023

PONE-D-21-32282R3 

Need for Affect, Need for Cognition, and the Desire for Independence 

Dear Dr. BARCELO SOLER:

I'm pleased to inform you that your manuscript has been deemed suitable for publication in PLOS ONE. Congratulations! Your manuscript is now with our production department. 

Kind regards, 

on behalf of

Dr. I-Ching Lee 

Academic Editor

PLOS ONE